# Mechanisms of Organizational Cultural Tightness on Work Engagement during the COVID-19 Pandemic: The Moderating Role of Transformational Leadership

**DOI:** 10.3390/bs13010027

**Published:** 2022-12-27

**Authors:** Xudong Song, Kan Shi, Wei Zhou

**Affiliations:** 1College of Education, Wenzhou University, Wenzhou 325035, China; 2Academy of Wenzhou Model Development, Wenzhou University, Wenzhou 325035, China; 3School of Economics and Management, University of Chinese Academy of Sciences, Beijing 100190, China

**Keywords:** organizational cultural tightness, transformational leadership, team-member exchange, work engagement

## Abstract

Cultural tightness–looseness, one of the cultural dimensions that play an essential role in organizational development, is changing the perception of psychology and behavior in organizations. This study conducted a paired questionnaire survey of leaders and their employees from five Chinese companies over three periods during the COVID-19 pandemic. The results found that organizational cultural tightness was more influenced by transformational leadership. Different from previous findings, in the context of the Chinese epidemic, organizational cultural tightness positively predicted employees’ work engagement with the moderating effect of transformational leadership. Team-member exchange also mediated employees’ work engagement, which had a facilitative effect on employees’ work engagement. In future research, the contingent effects of other leadership styles and organizational cultural tightness will be explored to reveal the different mechanisms of action on employees’ work engagement.

## 1. Introduction

Current research on organizational behavior in the occupational field has focused on negative organizational influences and countermeasures, such as psychological stress and job burnout on employees, with insufficient attention paid to positive psychological characteristics, such as employees’ work engagement [1]. Employees’ work engagement, as a positive factor measuring work attitudes, is strongly related to outcome variables such as organizational commitment [2], job performance [3], employee willingness to stay [4], and job satisfaction [5], and it is the opposite to the job burnout at the same continuum ends [6,7], with the influence of job resources being significant [8]. A related meta-analytic study noted that various job resources significantly impact work engagement [9]. However, such studies rely heavily on the Job Requirements-Resources Model (JD-R Model) for understanding and explaining employees’ work engagement. The existing JD-R Model should incorporate some new elements. We argue that tight and loose cultures, as an essential factor in the macro-cultural environment, should draw researchers’ attention and focus. Existing research frameworks ignore the antecedent role of organizational cultural factors on work engagement. The influence of hierarchical culture, team climate, and collectivist–individualist values in organizations has been explored in recent years [10]. We point out that whether and how organizational cultural tightness influences work engagement should be included in the research framework.

In recent years, organizational cultural tightness, based on the cultural tightness–looseness theory, has emerged as a new perspective for examining various organizational psycho-behavioral phenomena. In particular, the cultural differences between tight and loose cultures have been widely discussed during the COVID-19 pandemic [11]. Based on the perceived strength of social norms and tolerance for deviant behavior among organization members, we conducted an empirical investigation during the pandemic in China to examine the impact of organizational cultural tightness on work engagement in the particular periods. In addition, employees’ work engagement, as a category of research on the relationship between leadership and followership, is more often explored in current research in terms of “leader-centered” leadership style [12,13] and leader-member interaction [14] on employee engagement. While this may demonstrate the direct impact of leadership behaviors on employment, it also reflects the lack of evidence of the effects of organizational culture and leadership behaviors on employee engagement at the team level. To address these issues, we will explore the impact of organizational cultural tightness and team-member exchange on employees’ work engagement from the perspective of organizational culture and focus on the interaction of transformational leadership and organizational cultural tightness to produce the contingency changes.

## 2. Theoretical Background and Research Hypothesis

### 2.1. Organizational Cultural Tightness and Work Engagement

Organizational culture is defined as shared beliefs, perceptions, and expectations that characterize how an organization solves problems in a particular way [15]. Previous research has focused extensively on elements of organizational culture related to rules and order. For example, hierarchical culture is considered to be the most controlling type of culture among employees because of the strict rules and expected behaviors set among them [16] and it also has certain negative consequences on employee motivation and initiative [17]. In this kind of culture, employees generally act strictly according to the established management rules [18], thus limiting the employees’ proactive engagement and creativity in their work. This is a common perception, and studies have explored the impact of rules and order on work engagement in the context of related concepts such as “collectivism–individualism” and “hierarchy of power”. These concepts have their connotations and do not directly examine the impact of rules on employee engagement from the perspective of the perceived norms shared by organizational members.

The organizational cultural tightness explored in this study refers to the organization’s cultural strength. Pelto found that different early human communities, including the Eskimo, Cubeo, Hutterites, and Samburu, showed significant cultural differences between each other in terms of cultural tightness [19]. In recent years, Gelfand et al. have systematically proposed an academic concept of “tight and loose cultures” to distinguish the differences in cultural strength in modern countries, mainly based on the fact that different countries have different cultural tightness [20]. Tight cultures emphasize strong social norms and low tolerance for deviant behavior, whereas loose cultures emphasize weak social norms and high tolerance for deviant behavior [20]. Cultural tightness–looseness theory has its roots in several disciplines, including anthropology [19], sociology [21], and psychology [22]. Cross-cultural studies in 33 countries have shown that cultural tightness–looseness varies widely across the world and that this structure is completely distinct from cultural values in the traditional research sense [20]. According to the cultural tightness–looseness theory, tight and loose cultures are a systemic model from macro to micro, where macro social culture influences the psychological behavior of social institutions, organizations, teams, and individuals, where work institutions are open systems that perpetuate and reinforce dominant norms in the social environment, and where there are stable cultural looseness characteristics in the workplace and organizational team [20,23]. In contrast, at the work team level, organizational cultural tightness refers to the strength of the organizational norms shared by members in the work organization and the tolerance of deviant behavior by organizational members. Less existing research has explored the impact of organizational cultural tightness on individual psychological behavior in work organizations from this perspective.

What are the effects of organizational cultural tightness on employees’ work engagement in the workplace? There is little domestic and foreign literature on the influence of cultural tightness on work engagement, and the conclusions drawn based on the existing literature and data are not the same. Work engagement has been defined as a positive, fulfilling, work-related mental state characterized by energy, dedication, and concentration [24], and is a positive, fulfilling, work-related emotional and cognitive state in addition to indicating psychological identification with the job or organization [25,26]. Conceptually, in general, organizational cultural tightness has a negative impact on employees’ work engagement. In a meta-analysis, Rattrie et al. showed that a tight culture is more detrimental to employees’ work engagement than a loose culture [27]. At the same time, the study pointed out that high work demands are the main reason for the negative impact on work engagement in a tight culture. However, the existing studies are mainly based on the results of surveys and analyses in Europe and the United States, and the research periods are primarily in everyday situations. The same findings may not be applicable in countries with tight cultures. There is a lack of targeted research on whether similar results exist for organizational cultural tightness on employees’ work engagement. Taras et al. argue that organizational culture is a stronger predictor of individual employee work attitudes and behaviors in a tight culture than in a loose culture [28]. In theory, China, a representative country of a tight culture, will have a higher level of discipline and order in workplace organization. The tighter the organizational culture, the stronger organizational rules individuals perceive. The more restrictive individuals’ behavior within the organization will be unconscious, and subordinate employees will have difficulty engaging and creating at work [20]. In terms of relationships between team members, stronger perceptions of social norms can, to some extent, hinder the development of relationships between team members. It has been pointed out that the quality of relationships among team members in the organization is more of an informal relationship, and the atmosphere of the tight culture and low tolerance of transgressions by team members can also be detrimental to the development of good communication and social relationships among organizational members. This lack of interpersonal resources, essential supportive resources for work engagement, can lead to lower work engagement among employees in this state. However, it has also been shown that such findings do not apply in countries with tight cultures. For example, Turkish researchers found a significant positive predictive effect of cultural tightness on work engagement in their empirical data [29]. The adaptability of previous findings can also be influenced by specific survey contexts and significant events that have occurred historically. In this study, which was conducted mainly in China, a representative country with a tight culture, and during the COVID-19 pandemic, we suggest that leaders and employees in a tight culture are more likely to come together to face the crisis and challenges and also show a high level of work engagement in the face of the solid threat for life and health posed by the coronavirus. Accordingly, we argue that organizational cultural tightness will contribute to employee engagement.

### 2.2. The Mediating Role of Team-Member Exchange

Team-member exchange (TMX) refers to “individual members’ perceptions of their overall exchange relationships with other members in the team” [30]. It encompasses members’ perceptions of their willingness to help other members, share ideas and feedback, and receive information, help, and the extent of recognition from other members. The quality of TMX varies in both content and intensity. High-quality TMX relationships emphasizes social–emotional exchanges, implying more solidarity and assistance among team members [31], while low-quality TMX relationships are task-completion oriented, limiting team members’ interactions in accomplishing tasks and can lead to a lack of cooperation, communication, and trust among team members [32]. Theoretically, the creation of TMX changed the previous view of leaders and supervisors as the primary role. It shifted the previous vertical relationships in LMX to horizontal relationships, thus casting the perspective on the peer group at work. The vast majority of employees spend more time interacting with their peers than with their leaders during the work process, so the influence of peer groups may be more significant than that of leaders in some cases. Given that organizational cultural tightness is an emerging concept, little existing research directly addresses the relationship between the role of organizational cultural tightness on team-member exchange. This paper is dedicated to exploring how organizational cultural tightness affects the team-member exchange from the perspective of organizational cultural tightness.

In the research on the relationship between TMX and work engagement, the vast majority of the literature suggests that TMX positively impacts work engagement. First, high-quality TMX reflects the willingness of team members to be open and supportive of each other. This willingness leads to close psychological connections and effective working relationships with colleagues [30], enabling employees to engage in their work thoroughly. Second, high-quality TMX means that team members are more willing to share task-related resources, communicate and give feedback on work situations closely and immediately, and work together to solve problems or difficulties encountered at work, so as to promote employees’ intrinsic motivation and emotion to increase their work engagement [31]. Third, according to the job demands–resources model (JDR), job demands and resources can influence work engagement in an organizational setting [32]. High-quality team-member exchange implies more social interactions among employees, and this social interaction is a practical job resource that contributes to positive organizational outcomes [33]. Liao et al. showed that workplace horizontal social exchange relationships (i.e., TMX) are essential for work engagement [34]. Ancarani’s study in a public hospital showed that TMX could significantly and positively predict work engagement [35]. Vough et al. noted that interpersonal interactions among work peers could achieve their proactive goals and increase motivation and engagement at work [36].

Although TMX has not been studied as a mediating variable in the role of organizational cultural tightness and employees’ work engagement, existing studies have been conducted at the team level to provide evidence for the effect of team-member exchange on organizational culture and leadership behavior on team performance. For example, Murillo’s findings suggest that TMX as a mediating variable in team trust impacts collective effectiveness and team performance [37]. Ko’s study found that transformational leadership style and team members’ collectivism (collectivism) would help develop high-quality TMX among team members [38]. As a mediating variable, TMX enhances team social cohesion, performance, and viability. Although some research progress has been made, some of the most critical factors, including hierarchical culture, team climate, and collectivist–individualist values in organizations, have not received much attention in previous explorations of work engagement antecedents at the organizational team level. Research on rules and order in organizational culture has gradually attracted the interest of researchers in recent years, but the available evidence of relevant empirical studies is still scarce. Accordingly, we propose the concept of “organizational cultural tightness” that exists objectively in organizations to extend the existing “corpus” of research on the antecedents of work engagement in organizational culture.

### 2.3. The Moderating Role of Transformational Leadership

As a positive attitude of employees, work engagement is mainly seen as a manifestation of the relationship between leadership and followers, indicating the direct influence of leadership behaviors on employees’ work engagement. Although leadership behavior is an unavoidable topic of employees’ work engagement, it has also led to existing research exploring leadership styles more from the “leader-centered” perspective [12,13], leader–member interaction [14] and other influences on employees’ work engagement. Accordingly, leadership has always been placed as the primary position with the starting point for examining work engagement, but this solidified thinking does not always seem to be correct. Leadership is an influencing process based on the job demand–resource theory [39]. It can also be understood as a critical work resource influencing work engagement. Accordingly, we seek a shift in thinking to consider transformational leadership behavior more as a moderating factor.

Transformational leaders motivate their subordinates to sacrifice their interests for the sake of the organization by giving them the significance and value of the task they are undertaking so that their intrinsic motivation and higher-level needs are stimulated, and they can achieve results beyond their original expectations [40]. Numerous empirical studies have shown that transformational leadership can positively predict work engagement by increasing employees’ job resources [41], and researchers have been exploring the mechanisms of its impact [42]. Transformational leadership can serve as an accessible job resource. This resource’s acquisition will promote positive changes in employees’ work attitudes, accompanied by higher work engagement, higher job satisfaction and job performance, so that long-term organizational growth can progress steadily [25]. The interaction between leadership behavior and cultural factors is one of the focal points examined in this study. From an open systems perspective [43], leadership processes do not occur in a vacuum. They are influenced by the broader cultural values embedded in the organization [44,45]. Previous research has primarily examined the moderating role of organizational culture in the part of transformational leadership on organizational outcome variables. For example, Golden and Shriner discussed in a study whether organizational culture moderates the relationship between transformational leadership and employees’ self-rated creative performance. Their results found that different types of organizational culture play different degrees of moderating effect [46]. Transformational leadership also play a moderating role [47,48]. In China’s tight culture, transformational leadership behaviors can significantly moderate the relationships among the team, thereby better-aligning team members to work together. This suggests that a good organizational climate and team-member exchange will facilitate the generation of employee engagement. Therefore, we examine the moderating role of transformational leadership in the influence of organizational cultural tightness on the work engagement from the perspective of organizational cultural tightness.

### 2.4. General Hypothesis Model

In terms of the above literature review and analysis, the research is based on the cultural tightness–looseness theory [20], which views employees’ work engagement as a systematic motivational process. The hypothetical model of the study, shown in Figure 1, essentially explores how organizational cultural tightness and leadership behaviors jointly influence team-member exchange and thus promote employees’ work engagement. We argue that due to the threat of the COVID-19 pandemic, China, as a tight culture, will generate a series of policy measures when faced with a crisis, which will also affect the psychological behavior of individuals in the work organization. While organizational cultural tightness may inhibit employee engagement under conventional conditions, among the countries with tight culture in the crisis situation of the epidemic, organizational cultural tightness may promote engagement to help the organization get through the situation faster. Transformational leadership can act as a motivational incentive. In addition, team-member exchange, a critical factor contributing to work engagement, was identified as a key mediating variable in this study. Accordingly, based on the above literature and analysis, the following hypotheses were formulated.

**H1:** 
*Organizational cultural tightness has a positive predictive effect on employees’ work engagement.*


**H2:** 
*Team-member exchange plays a mediating role between organizational cultural tightness and employees’ work engagement.*


**H3:** 
*Transformational leadership positively moderates the predictive role of organizational cultural tightness on team-member exchange.*


## 3. Methodology

### 3.1. Participants and Procedures

In this study, we conducted a paired questionnaire survey of the leadership and employee teams of five private enterprises in Zhejiang Province, China, which are mainly engaged in the production and operation activities of garments, electrical appliances, and mechanical chemicals, and are medium-sized enterprises in the manufacturing category in terms of the scale of their activities. After gaining the approval and support of the top leaders of the companies for the survey, the research was actively coordinated by the HR managers of the companies. A random sampling method was used to draw participants within the company, and the model of matching leaders and employees up and down in the enterprises was used, where each leader below the middle level (team leader) was compared with five randomly selected employees. The paper-based survey conducted a questionnaire survey. Two sets of targeted questionnaires were designed for business leaders and employees. A combination of self-assessment and other assessments was used to collect research data. To ensure that the data were representative, we randomly selected five employees from each team to participate in the survey. We coded each team’s leaders and affiliated employees to ensure that the results were confidential. In the beginning, 75 questionnaires for leaders and 375 for employees were distributed. Then, the survey was conducted in three phases, each phase of the questionnaire lasting one week. In the first data collection period 1 (T1), employees filled out data on demographic variables, the organizational cultural tightness Questionnaire, and the Transformational Leadership Questionnaire. Two months later, in period 2 (T2) of the second data collection, the second batch of questionnaires was given to employees who completed the questionnaires in T1 only, allowing them to complete the work engagement questionnaire. A final data collection was conducted two months later in period 3 (T3) to give questionnaires to employees’ leaders to achieve the employees’ work engagement questionnaire. Since there was some attrition of both leaders and employees in each stage, 51 leaders and 255 paired subordinates were collected, totaling 306 people (68.00% data efficiency). After data cleaning to eliminate invalid data, 228 valid data were finally obtained (74.51% data efficiency). Among them, 136 (59.65%) were male, and 92 (40.359%) were female, with an average age of 39.39 (*SD* = 8.762). The ethics committee of Wenzhou University reviewed our research, and all subjects provided informed consent and confirmation of this survey before receiving the questionnaire.

### 3.2. Measurement

#### 3.2.1. Organizational Cultural Tightness Scale (T1)

The study was adapted from a scale developed by Gelfand et al. [20] to investigate cultural tightness–looseness in a broad sense. The scale includes six items, which are the number and clarity of organization norms, tolerance of rule violations, and overall compliance with organization norms in each team, such as the psychological scoring agreement with the statement, “I feel that if someone in our team misbehaves, others will strongly disagree”. The “organization norms” mentioned in the scale usually refer to the unwritten standards of behavior in the team. The scale was scored on a 6-point Likert scale from “totally disagree” to “totally agree”. The Cronbach’s alpha coefficient in the study for the entire scale was 0.91 and the MacDonald’s omega coefficient was 0.91.

#### 3.2.2. Transformational Leadership Scale (T1)

The Transformational Leadership Questionnaire (TLQ), developed by Chao-Ping Li and Kan Shi [49], was used in the research. The TLQ consists of 26 questions, which refer to four dimensions: “moral leadership”, “visionary motivation”, “leadership charisma”, and “personalized care”. Based on the principle of model simplification, we packaged the questions of each of the four dimensions. Real effective motivation and communication are critical factors in enhancing innovation. How to achieve effective motivation and communication in the Chinese context is given special consideration in our study. The commonly used measurement questionnaire in the Western context is the MLQ (Multifactor Leadership Questionnaire) developed by Bass [39]. Still, more and more scholars have found that the fit index and reliability of the MLQ in the Chinese context are unsatisfactory [50]. According to this, we chose TLQ for this research. The Cronbach’s alpha coefficient in the study for the entire scale was 0.79 and the MacDonald’s omega coefficient was 0.86.

#### 3.2.3. Team-Member Exchange Scale (T2)

An adapted version of Seers et al.’s [51] 10-item scale was used to rate TMX quality during our research. The original Cronbach’s alpha was 0.88 in terms of the preferences of bank managers [52]. The scale consists of 10 question items and is scored on a 5-point Likert scale. Representative items of the scale include: “ How often do you make suggestions to other team members about better ways to do your job?” and “How often do you make a voluntary effort to help others on your team in complicated situations?”. In the study, the Cronbach’s alpha coefficient for the entire scale was 0.94 and the MacDonald’s omega coefficient was 0.94.

#### 3.2.4. Employees’ Work Engagement Scale (T3)

The employees’ work engagement scale developed by Schaufeli et al. [24] was used. There are 14 items total, including three sub-dimensions of energy, dedication, and concentration. A 5-point Likert scale was used, with one representing “strongly disagree” and five representing “strongly agree”. The examples were that “I feel strong and energetic when I work”, “I am passionate about my work”, and “When I am working, time always flies”. The Cronbach’s alpha coefficient in the research for the entire scale was 0.90 and the MacDonald’s omega coefficient was 0.93. 

### 3.3. Data Analysis

We used SPSS 24.0 and Mplus 8.3 statistical software for correlation analysis, regression methods, and structural equation modeling (SEM) development and validation of the variables in the study. The least squares-based operations were used to estimate direct and indirect effects. The CINTERVAL (BCboostrap) test was used to estimate path significance testing and analysis of effect sizes. 

By reliably estimating the reliability of each variable in the study, we calculated the Cronbach’s alpha coefficient and MacDonald’s omega coefficient for each variable data through validated factor analysis of the measurement model. It was found that the Cronbach’s alpha coefficient and MacDonald’s omega coefficient for organizational cultural tightness (T1) and team-member exchange (T2) were consistent, and the MacDonald’s omega coefficient for transformational leadership (T1) and employee work engagement (T3) was higher than the Cronbach’s alpha coefficient. By comparing the magnitudes of the two and combining the conditions for the potential hypothesis of Cronbach’s alpha coefficient to be satisfied [53], it can be inferred that the two variables of transformational leadership (T1) and employees’ work engagement (T3) in the research most likely do not satisfy the tau-equivalence assumption [54], causing the Cronbach’s alpha values appear to underestimate the reliability [55,56,57]. Therefore, in our study, MacDonald’s omega coefficient may be more scientific [58]. From the results, the reliability index of the data of each variable is higher than 0.7 for both Cronbach’s alpha coefficient and MacDonald’s omega coefficient, indicating that the reliability of the data of each variable meets the measurement criteria [59].

## 4. Results

### 4.1. Test for Common Method Bias

The common deviation analysis was performed on all valid data using Haman’s one-way analysis of variance. The results showed that 21 factors had root values greater than 1, with the first factor having a variance of 31.008%, which is less than the 40% threshold. We also performed a CFA on the one-factor model, which showed a poor fit (*χ*^2^/*df* = 9.25, CFI = 0.59, TLI = 0.58, NFI = 0.56, RFI = 0.55, RMSEA = 0.09). According to the CFA method, a poor fit index of the one-factor CFA model indicates that the common method bias is not severe. Therefore, the common method bias of the data in this study was not significant.

### 4.2. Correlation Analysis

The descriptive statistics between the variables and the Pearson product difference correlation analysis table are shown in Table 1.

The scatter plot revealing the two-by-two correlations between the variables is shown in Figure 2. From the obtained result plot, we can quickly and visually find that the four variables involved in the study, organizational cultural tightness, transformational leadership, team-member exchange, and employees’ work engagement, show positive correlations between each of the two. From the slope of the fitted linear trend line and the correlation coefficients in Table 1, we can find that organizational cultural tightness is positively correlated with transformational leadership (*r* = 0.48, *p* < 0.01), team-member exchange (*r* = 0.46, *p* < 0.01), employees’ work engagement (*r* = 0.12, *p* < 0.05), transformational leadership with team-member exchange (*r* = 0.58, *p* < 0.01), employees’ work engagement (*r* = 0.20, *p* < 0.01), and team-member exchange with employees’ work engagement (*r* = 0.31, *p* < 0.01). The positive correlations are significant and provide the basis for further linear regression analysis and structural equation modeling later on.

### 4.3. The Mediating Effect Model Test

To explore the mediating effect (indirect effect) of team-member exchange, our study conducted path regression and tests of mediating effects (as shown in Table 2). The 95% confidence test was conducted by performing CINTERVAL (BCboostrap) test in Mplus with 1000 iterations of sampling in the sample data, and the estimation method was the Maximum Likelihood Estimate (MLE). As shown in Table 2, after controlling for gender, age, and education of leaders and employees, the results indicated that organizational cultural tightness was a significant positive predictor of work engagement (b = 0.394, Z = 3.007, *p* < 0.01), and Hypothesis 1 was supported. In the path analysis, the positive predictive effects of organizational cultural tightness on team-member exchange (*b* = 0.414, *Z* = 4.208, *p* < 0.001) and team-member exchange on work engagement were significant (*b* = 0.332, *Z* = 2.366, *p* < 0.05), and further the mediating effect of team-member exchange in the model, the results indicated that the mediating effect of “organizational cultural tightness → team-member exchange → work engagement” in the model was significant (*b* = 0.13, *Z* = 2.12, *p* < 0.05), and the hypothesis 2 was supported by the data.

### 4.4. The Moderated Mediation Model Test

Organizational cultural tightness is an organizational-level variable whose effect on team-member exchange is moderated by transformational leadership. Therefore, we further conducted a moderating analysis on whether transformational leadership behavior plays a role in the path of organizational cultural tightness, using a robust Maximum Likelihood Robust Estimator (MLR) with a fit index (Convergence) of 0.01. As shown in Table 3, after controlling for gender, age, and education of leaders and employees, moderated mediating effects were tested for transformational leadership between tight and loose cultures and team-member exchange. The results indicated that the cross-product term between organizational cultural tightness and transformational leadership was significant (*b* = 0.083, *Z* = 2.233, *p* < 0.05). These results suggest that transformational leadership plays a significant positive moderating role. Hypothesis 3 of the study was supported.

Further simple slope analysis shows that, as shown in Figure 3, the contribution of organizational cultural tightness to work engagement is significantly enhanced by adding different levels of transformational leaders. This suggests that in the context of our tight culture, transformational leadership behaviors can significantly and positively moderate the entire team-member exchange relationship and thus better align team members to work together and engage in their work.

## 5. Discussion

Based on the cultural tightness–looseness theory, the data results obtained from the survey analysis and the domestic and international research literature, we explore the impact of organizational cultural tightness on employees’ work engagement and, in the process, discuss the mediating role of team-member exchange in it and the moderating role of transformational leadership. The following main theoretical contributions are as follows. 

First, we incorporate organizational cultural tightness as an antecedent variable of employees’ work engagement. It obtains evidence of the existence of a facilitative effect of organizational cultural tightness on employees’ work engagement through path analysis. The findings obtained in this study differ from previous studies that concluded that organizational cultural tightness has a negative effect on employee engagement [27]. Our interpretation of the new findings obtained is mainly based on a new perspective of cultural tightness–looseness theory and a comprehensive evaluation of our findings from the perspective of the characteristics of the Chinese national culture and the working conditions of the participants’ companies. The investigation was conducted in the context of a major historical event, the COVID-19 pandemic, where China, as a representative country of a tight culture, tends to adopt a highly synchronized and committed attitude to address the challenges due to its greater sensitivity to crises historically and ecologically relative to countries with loose cultures [20]. An epidemic such as the COVID-19 pandemic would trigger a high priority, and strict epidemic prevention measures would be implemented in China. This also affects teams and individuals within work organizations. In the broader context of epidemic prevention and control in China. Moreover, China’s traditional national culture has a continuous history of more than 5000 years, which makes the power of history and social culture have a strong influence on the organization and individual employees. The “ritual” culture of traditional Chinese culture represents the high demand of individuals for their own moral code, and the “family and nation” culture essentially represents the tendency of individuals to assume the responsibility of family and nation in the face of major crisis events, to unite and resist difficulties together. Under the influence of such a national culture and individual interaction, organizations tend to set high organizational norms and low tolerance for individual deviations within the organization, including stricter organizational order and more supervision. In the general environment of epidemic prevention and control in China, the tight state of organizational culture is activated, which affects teams and individuals in the work organization, and leaders and employees within the organization shift to the psychological state in crisis situations, which is manifested by increasing the intensity of organizational norms to stimulate the engagement state of employees. Furthermore, the main participants in this study were groups of leaders and employees from private firms, and in a comparative study of firms with different ownership, transformational leadership style was identified as a major factor influencing the strength of organizational culture, and the interrelationship between leaders and employees was stronger in private firms [60]. In the face of the potent threat to life and health posed by the coronavirus, leaders and employees in the organization of tight culture tend to come together to face the crisis and challenges and increase the cultural tightness of the organization as a whole, which to some extent also leads to a sense of consistency and identity in organizational goals, so as to trigger employees to generate a higher level of work engagement status in the pandemic. The study extends the application area of cultural tightness–looseness theory to the study of work engagement. In addition, it interprets the results in the context of the characteristics of Chinese national culture and the changes in organizational work status of companies before and after the epidemic, providing a perspective for better understanding of employees’ work engagement under different cultural tightness and looseness. In addition, the existing literature on the understanding and interpretation of work engagement relies heavily on the traditional job demands–resources model, while the model is also evolving [10]. The organizational cultural tightness involved in this study is an essential addition to work engagement theory research. In summary, we enrich the literature supporting the organizational cultural tightness factor in previous work engagement studies and analyze and explain the findings obtained through the cultural tightness–looseness theory.

Second, we highlight the mediating role of team-member exchange in the effect of organizational cultural tightness on work engagement. First of all, the present study draws a different conclusion on the effect of organizational cultural tightness on TMX than previous research, which suggests that high-quality TMX relationships are usually born in a safe, open, and positive interpersonal and psychological environment [34]. However, the results obtained in this study contradict this by adopting the perspective of increased team member unity due to the “adaptability” of the tight culture theory [20]. High organizational norms, monitoring, and punishment reduce intra-organizational variation in a tight culture work organization, resulting in higher cultural strength and coherence. In the face of crises and challenges posed by COVID-19, the power of this tight culture enhances high synchronization and consistency in team members’ working relationships [61]. This consistency leads to higher quality TMX within the organization, as evidenced by a high level of consistency and mutually supportive cognitive states among team members, which strengthens close psychological ties and effective working relationships [30], enhancing the quality and level of work engagement [62]. Moreover, from the perspective of reciprocity, this high level of consistency and synchronization also leads to the establishment of higher reciprocal bargaining among members. Suppose members’ contributions are recognized and rewarded by others in the team. In that case, this will contribute to the organization’s achievement of shared team goals and lead to higher levels of work engagement in this climate. Moreover, the team member relationships examined in this study are based on the “horizontal” relationships between team members, as opposed to the “vertical” relationships between leaders and employees that have dominated the literature [12]. To a certain extent, this is a break from the “leader”-centered approach in previous studies. Lastly, according to the cultural tightness–looseness theory, all participants in this study came from China, a representative country of tight culture. As countries with tight culture always face more challenges in dealing with external environment and real threats [20], the working team under the cultural background of China has a natural tendency of cultural tightness. Therefore, team-member exchange in Chinese work organizations may be a key mediator in explaining the positive effects of organizational cultural tightness on employees’ work engagement in the face of crises and challenges posed by the COVID-19 pandemic.

Third, we reveal that organizational cultural tightness plays a positive moderating role in promoting organizational employees’ work engagement by transformational leaders, which to some extent, enriches the theoretical framework of the interaction between leadership behavior and organizational cultural factors. Leadership processes do not occur in a vacuum [43], and the influence of leadership behavior and the social culture in which the organization is embedded must be considered simultaneously [44,45]. Therefore, during the research, in examining the mechanism of the effect of organizational cultural tightness on work engagement, since leadership behavior is one of the powerful means to eliminate the unfavorable components of cultural differences [63], we also focused on the influence of transformational leadership style to obtain the best motivational effect on employees’ work engagement. It was shown that transformational leadership could essentially motivate employees’ work engagement by articulating a vision, virtue modeling, etc., similar to the findings obtained from previous studies that transformational leadership creates a good team atmosphere that promotes collaboration, mutual trust, and cooperation [64]. From the perspective of social exchange theory and resource conservation theory, transformational leaders facilitate reciprocity and exchange among team members and give help to other team members based on the resources they have, which is reflected in collaboration, mutual trust and loyalty, cooperation, and social rewards among team members. The unique factor of transformational leadership in the Chinese context, the “virtuous example,” addressed in this study, can motivate employees to engage in their work. Furthermore, transformational leadership can be seen as an immediate work resource available to employees. As this perceived work resource increases, a quality exchange relationship is formed between employees, not just a monetary one, which means that the quality of exchange between team members increases. This suggests that transformational leadership can reinforce the effect of organizational cultural tightness in facilitating team members’ exchange and work engagement.

Finally, we also provide three practical implications for the motivational process of increasing the level of employees’ work engagement in human resource management. Firstly, in a company’s innovation management, the cultural tightness in the organization of each innovation team becomes an essential factor to be considered. The current situation can be determined by measuring the existing cultural tightness. The cultural tightness of the work teams in the company can be adjusted to a certain extent dynamically by enacting and changing the corresponding regulations according to the vision and mission of the work teams in the company. Secondly, based on the perspective of tight-loose cultures, business leaders can combine their behavioral leadership styles with a cultural tightness–looseness contingent management strategy. By strengthening norms in some areas and relaxing them in others, it can ensure, to a certain extent, that the tight and loose cultures become a solid and beneficial force that plays a vital role in leadership behavior in terms of positive attitudes toward employees’ work. Thirdly, the team-member exchange is a crucial factor to be considered in the company’s management. The quality of the relationship between team members may be one of the key factors in motivating employees to work. By facilitating interactions between team members in a company, employees are more likely to enter a state of work engagement, promoting employee performance.

## 6. Research Gaps and Future Directions

Although some significant findings are yielded in the research, future studies still need to refine certain research limitations further. On the one hand, the role of organizational culture tightness–looseness and transformational leadership on employees’ work engagement can be further explored by controlling for team members’ values (e.g., individualism–collectivism), which may lead to more convincing findings. On the other hand, the research framework on the influence of organizational culture and the contingent effect of leadership behavior will be supplemented with additional leadership behavior styles (transactional leadership, etc.) to be analyzed and compared with cultural tightness–looseness to synthesize the impact on employees’ work engagement.

## 7. Conclusions

Based on the cultural tightness–looseness theory, we explored the effects and mechanisms of organizational cultural tightness on employees’ work engagement through an empirical investigation of Chinese business organizations during the COVID-19. The results show a significant positive predictive effect of organizational cultural tightness on employees’ work engagement, in which team-member exchange plays a mediating role. The mediating effect indicates that the higher the quality of team-member exchange, the higher the level of employees’ work engagement. In addition, the interaction between transformational leadership and organizational cultural tightness is significant and positively moderates. The moderating effect suggests that transformational leadership enhances the positive effect of organizational cultural tightness on employees’ work engagement. We also provide evidence and explanations for the positive effect of organizational cultural tightness on work engagement in order to offer feasible solutions for motivating and enhancing the level of work engagement of employees in the organization. 

## Figures and Tables

**Figure 1 behavsci-13-00027-f001:**
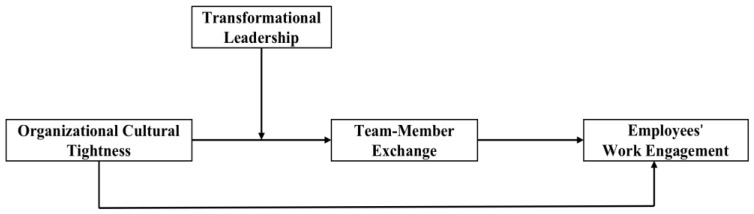
Conceptual model.

**Figure 2 behavsci-13-00027-f002:**
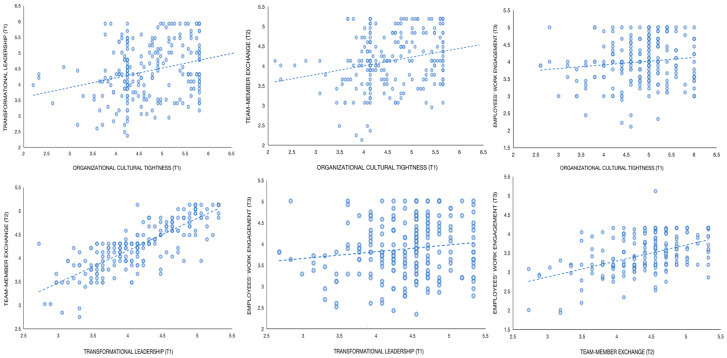
Scatter plot between variables.

**Figure 3 behavsci-13-00027-f003:**
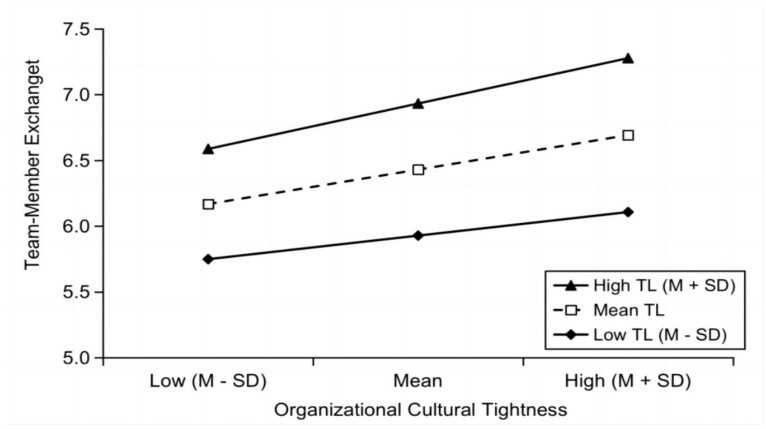
Simple slope analysis diagram.

**Table 1 behavsci-13-00027-t001:** Correlation Analysis Results Between Variables.

Variable	*M*	*SD*	1	2	3	4	5	6	7	8	9	10
1. Employee Gender	0.65	0.48	1									
2. Employee Age	38.59	9.14	0.19 **	1								
3. Employee Education	1.75	0.80	−0.44 **	−0.21 **	1							
4. Leader Gender	0.70	0.46	0.55 **	0.14 *	−0.39 **	1						
5. Leader Age	42.62	6.03	0.045	0.20 **	0.14 *	0.10	1					
6. Leader Education	2.21	0.82	−0.45 **	−0.11	0.36 **	−0.50 **	−0.08	1				
7. Team Cultural Tightness	5.01	0.70	0.08	0.30 **	−0.13	0.16 *	0.09	−0.17 *	1			
8. Transformational Leadership	4.21	0.52	0.11	0.14 *	0.01	0.14	0.18 **	−0.12	0.48 **	1		
9. Team-Member Exchange	4.14	0.55	0.19 **	0.19 **	−0.08	0.26 **	0.12	−0.25 **	0.46 **	0.58 **	1	
10. Work Engagement	3.97	0.65	0.21 **	0.12	−0.14 *	0.39 **	0.12	−0.23 **	0.12 *	0.20 **	0.31 **	1

Note: * means *p* < 0.05; ** means *p* < 0.01.

**Table 2 behavsci-13-00027-t002:** Model’s Regression Path and Mediation Effect Analysis.

Variable	IV: Organizational Cultural Tightness (T1)			Mediator: Team-Member Exchange (T2)			DV: Employees’ Work Engagement (T3)		
	*b*	*SE*	*Z*	*b*	*SE*	*Z*	*b*	*SE*	*Z*
Employee Gender	−0.21	0.12	−1.75	0.15	0.10	1.44	−0.08	0.11	−0.70 ***
Employee Age	0.02	0.01	3.11 **	0.00	0.00	0.35	0.00	0.01	0.56
Employee Education	−0.05	0.07	−0.69	0.07	0.05	1.53	−0.01	0.06	−0.13
Leader Gender	0.17	0.13	1.29	0.10	0.11	0.86	0.46	0.12	4.00
Leader Age	−0.00	0.01	−0.44	0.01	0.01	1.17	0.01	0.01	0.74
Leader Education	−0.11	0.06	−1.99 *	−0.05	0.05	−1.08	−0.02	0.06	−0.27
Organizational Cultural Tightness (T1)				0.41	0.10	4.21 ***	0.39	0.13	3.01 *
Team-Member Exchange (T2)							0.33	0.14	2.37 *
*R^2^*	0.15			0.32			0.23		
*F*	3.12 **			4.36 ***			4.22 ***		

Note: * means *p* < 0.05; ** means *p* < 0.01; *** means *p* < 0.001. For gender, 0 = female, 1 = male. T1/2/3 = Time 1/2/3.

**Table 3 behavsci-13-00027-t003:** Moderated Mediation Analysis.

Variable	IV: Organizational Cultural Tightness (T1)			Mediator: Team-Member Exchange (T2)			DV: Employees’ Work Engagement (T3)		
	*b*	*SE*	*Z*	*b*	*SE*	*Z*	*b*	*SE*	*Z*
Employee Gender	−0.17	0.10	−1.75	0.11	0.09	1.24	−0.07	0.09	−0.76
Employee Age	0.31	0.07	4.49 ***	0.04	0.07	0.63	0.04	0.08	0.49
Employee Education	−0.07	0.09	−0.77	0.07	0.07	1.17	−0.01	0.08	−0.16
Leader Gender	0.14	0.10	1.43	0.10	0.09	1.07	0.36	0.09	4.15 ***
Leader Age	−0.02	0.07	−0.25	−0.00	0.06	−0.02	0.04	0.07	0.64
Leader Education	−0.12	0.08	−1.39	−0.06	0.07	−0.84	−0.03	0.08	−0.42
Organizational Cultural Tightness (T1)				0.26	0.07	3.73 ***	0.39	0.11	3.49 **
Transformational Leadership (T1)				0.50	0.10	5.19 ***			
Organizational Cultural Tightness (T1) × Transformational Leadership (T1)				0.08	0.04	2.23 *			
Team-Member Exchange (T2)							0.27	0.10	2.79 **
*R^2^*	0.18			0.48			0.22		
*F*	2.88 **			7.24 ***			4.03 ***		

Note: * means *p* < 0.05; ** means *p* < 0.01; *** means *p* < 0.001. For gender, 0 = female, 1 = male. T1/2/3 = Time 1/2/3.

## Data Availability

The data that support the findings of this study are available from the corresponding author upon reasonable request.

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
