# Peer review of "Mechanisms of Organizational Cultural Tightness on Work Engagement during the COVID-19 Pandemic: The Moderating Role of Transformational Leadership"

_behavsci, 2022, doi:10.3390/bs13010027_

Round 1

Reviewer 1 Report

The paper deals with an original problem in the field of organizational behavior. The structure of the work is correct, logical and coherent. However, some issues still require rethinking and correction. The main reservations concern the research methodology. Hypothesis 2 needs to be changed. In its current form, it is incorrect. The term "partially" used is inaccurate. It is not possible to support such a hypothesis. There is also no explanation for the sampling selection of the surveyed companies. Why these 5 companies from the region were selected. What do these companies do? What is the scale of their activities? In the context of these questions, there is a doubt about the discussion and conclusions in the paper. To what extent are the presented research results universal, i.e. they can have a worldwide reference. The author also does not attempt to evaluate the results of the research from the perspective of the characteristics of the Chinese national culture and the work conditions related to it. The literature review in the field of cultural tightness of work also needs to be extended.

Reviewer 2 Report

In this paper, through surveys of employees and leaders, the effect of cultural rigidity during covid-19 in China is studied. The results found that organisational cultural rigidity was more influenced by transformational leadership than in the pre-covid-19 periods.

The work is well written and motivated. It is of relative importance, more within the geographical environment studied than outside the boundaries of that environment. Furthermore, 36.5% of the references used are articles published in the last 10 years and 23% in the last few years, demonstrating the topicality of the problem studied.

However, there are some points in the work that should be clarified:

1) Cronbach (1951) proposes the coefficient alpha (α) which estimates the proportion of variance of a measurement instrument due to the common factor between items. And it is advisable for researchers to take into account the assessment of compliance with its basic assumptions:

a) The so-called tau-equivalence which is that items measure the same trait or the same latent variable with the same or similar degree of accuracy (Cho, 2016).

b) Uncorrelation of errors as they are assumed to be independent (the error score of any pair of items is uncorrelated).

c) Unidimensionality, i.e., all items or questions must measure a single latent trait.

d) The construct measure must be continuous.

The tau-equivalence assumption assumes the same true score for all scale items or factor loadings of all equal items in a factor model and is a requirement for α to be unbiased (Cronbach, 1951). If the tau-equivalence assumption is not met, the value of Cronbach's alpha tends to underestimate (underestimate) the reliability of the test score. And, if the assumption of no correlation between errors is violated then the Cronbach's alpha value tends to overestimate the reliability value (Cho and Kim, 2015; Lucke, 2005; Raykov and Marcoulides, 2017).

Therefore, if the tau-equivalent measurement model does not hold true then other coefficients may be a better alternative such as the omega coefficient. When tau-equivalence exists then Cronbach's alpha and MacDonald's omega values coincide or are very close (Trizano-Hermosilla and Alvarado, 2016).

The authors should point out McDonald's omega and compare it with Cronbach's alpha.

b) Regarding the correlation matrix, I would have liked to see both the correlogram of each of the variables, as well as the different scatter plots, which are always very illustrative and informative.

Authors should represent and explain them in order to improve the results obtained.

 References

Cronbach, L.J. Coefficient alpha and the internal structure of tests. Psychometrika 16, 297–334 (1951). https://doi.org/10.1007/BF02310555

Cho E., Kim S. (2015). Cronbach’s coefficient alpha: Well known but poorly understood. Organizational Research Methods, 18(2), 207–230.

Cho, E. (2016). Making Reliability Reliable: A Systematic Approach to Reliability Coefficients. Organizational Research Methods, 19(4), 651–682. https://doi.org/10.1177/1094428116656239

Lucke J. F. (2005). “Rassling the hog”: The influence of correlated item error on internal consistency, classical reliability and congeneric reliability. Applied Psychological Measurement, 29(2), 106–125.

Raykov, T., & Marcoulides, G. A. (2017). Evaluation of true criterion validity for unidimensional multicomponent measuring instruments in longitudinal studies. Structural Equation Modeling, 24(4), 599–608. https://doi.org/10.1080/10705511.2016.1172486

Trizano-Hermosilla, I., & Alvarado, J. M. (2016). Best alternatives to Cronbach's alpha reliability in realistic conditions: Congeneric and asymmetrical measurements. Frontiers in Psychology, 7, Article 769. https://doi.org/10.3389/fpsyg.2016.00769
